

# Life history stage explains behavior in a social network before and during the early breeding season in a cooperatively breeding bird

Angela Tringali[1], David L. Sherer[1,2], Jillian Cosgrove[3] and Reed Bowman[1]

[1] Avian Ecology Program, Archbold Biological Station, Venus, FL, United States of America
[2] Department of Biology, University of Central Florida, Orlando, FL, United States of America
[3] Department of Fisheries and Wildlife, Oregon State University, Corvallis, OR, United States of America

## ABSTRACT

In species with stage-structured populations selection pressures may vary between different life history stages and result in stage-specific behaviors. We use life history stage to explain variation in the pre and early breeding season social behavior of a cooperatively breeding bird, the Florida scrub-jay (*Aphelocoma coerulescens*) using social network analysis. Life history stage explains much of the variation we observed in social network position. These differences are consistent with nearly 50 years of natural history observations and generally conform to *a priori* predictions about how individuals in different stages should behave to maximize their individual fitness. Where the results from the social network analysis differ from the *a priori* predictions suggest that social interactions between members of different groups are more important for breeders than previously thought. Our results emphasize the importance of accounting for life history stage in studies of individual social behavior.

## INTRODUCTION

The social environment can have far reaching consequences for individual survival and reproduction, and the magnitude of these effects may vary with age and sex (*Alberts, 2019*). In stage-structured populations, life history stages are not necessarily related to age and each is described by its own set of demographic parameters. Because selection pressures in each stage are different (*Schluter, Price & Rowe, 1991*; *Pujolar et al., 2015*), the stage-structure of a population can have profound impacts on its demography (*Tuljapurkar & Caswell, 1997*; *Caswell & Vindenes, 2018*), ecology (*Miller & Rudolf, 2011*; *Wesner, 2019*), conservation (*Crouse, Crowder & Caswell, 1987*; *Kindsvater et al., 2016*; *Van Rees et al., 2018*), and ability to respond to environmental change (*Cotto et al., 2019*).

The ability to quantify the social environment as it is experienced by each individual has led to improvements in the description of social structures (*Wittemyer, Douglas-Hamilton & Getz, 2005*; *Lusseau et al., 2006*; *Wey et al., 2008*; *Webber & Vander Wal, 2019*), insights

Corresponding author
Angela Tringali,
atringali@archbold-station.org

into individual variation and its consequences (*Oh & Badyaev, 2010*; *Aplin et al., 2013*), and a better understanding of interactions between the social and physical environment (*Firth & Sheldon, 2015*; *Pinter-Wollman, 2015*; *Leu et al., 2016*). The influence of individual traits, including stress physiology (*Boogert, Farine & Spencer, 2014*; *Moyers et al., 2018*), genetic relatedness (*Ilany & Akçay, 2016*), and personality (*Krause, James & Croft, 2010*; *Wilson et al., 2013*; *Snijders et al., 2014*; *Sih et al., 2018*) on social network position has been widely studied. Although several studies have examined the effects of reproductive status on social behavior (*Fischhoff et al., 2009*; *Patriquin et al., 2010*; *Wey et al., 2013*; *Menz et al., 2017*), few have explicitly examined the effects of life history stage (but see *Wey et al., 2013*).

Individuals in different life history stages have different strategies to maximize their fitness, which can affect social behaviors within a population (*Rudolf, 2007*; *Blumstein, Wey & Tang, 2009*; *Fischhoff et al., 2009*). Cooperative breeders are highly social and tend to have stage-structured populations (*Ekman et al., 2004*), making them excellent model systems in which to study the effects of life history stage on social behavior. Here, we examine the influence of life history stage on social behavior using social network analysis. In social networks, individuals are considered connected when they interact or are detected together (*Whitehead & Dufault, 1999*; *Krause, Lusseau & James, 2009*; *Farine, 2015*). Social network analyses are powerful because they move beyond dyadic interactions to quantify the social structure of groups or populations (*Croft, James & Krause, 2008*; *Cantor et al., 2019*). From these networks, a variety of metrics can be calculated, characterizing an individual's number and strength of connections as well as its position relative to others in the network (*Krause, Lusseau & James, 2009*).

Florida scrub-jays (*Aphelocoma coerulescens*) are a well-studied cooperative breeder, with thoroughly described stage-specific behavioral differences (*Woolfenden & Fitzpatrick, 1984*; *Woolfenden & Fitzpatrick, 1990*). They are territorial and non-migratory, living in family groups consisting of a single, monogamous breeding pair that monopolizes all reproductive effort and 0–7 helpers (*Woolfenden & Fitzpatrick, 1984*; *Townsend et al., 2011*). Helpers are most commonly the offspring of the breeding pair on whose territory they reside, but occasionally (less than 15%) are associated with unrelated or distantly-related breeders (*Woolfenden & Fitzpatrick, 1990*). Within these family groups a strict dominance hierarchy exists; breeders are the most dominant, and among helpers, males are dominant over females, and older birds are dominant to younger birds, and hierarchies exist among brood mates (*Woolfenden & Fitzpatrick, 1977*; *Woolfenden & Fitzpatrick, 1984*; *Tringali & Bowman, 2012*). Although this species is territorial, social relationships extend beyond territory boundaries. Dispersal distances are short, on average individuals only breed one or two territories away from their natal territory. This results in related jays tending to be clustered together on the landscape and allows for social relationships among parents, offspring, and siblings to persist after dispersal. Juveniles and helpers also make forays away from their natal territories, sometimes forming large, temporary aggregations of non-breeders. Dominance hierarchies also exist across cohorts, between jays from different family groups (*Woolfenden & Fitzpatrick, 1977*).

Florida scrub-jays' social system allows us to classify all adults into one of three life history stages: (1) breeders, who own territories and have been associated with a nest

with contents; (2) dominants, who own territories, but have not yet bred; and (3) helpers, who neither own territories nor breed (*Woolfenden & Fitzpatrick, 1984*; *Woolfenden & Fitzpatrick, 1990*). Typically, dominants have only recently acquired a territory, and may or may not have paired with a mate. Individuals usually remain classified as dominants for less than a year but in rare cases, dominant birds or their mates are infertile, failing to produce eggs after multiple breeding seasons. Hereafter, we will use dominant as it refers to life history stage, and not as a position within the social hierarchy.

Life history stage-specific behaviors have been described. Breeders defend their territories, engage in courtship behavior with their mate, and build and tend to their nest (*Stallcup & Woolfenden, 1978*; *Woolfenden & Fitzpatrick, 1984*). Both parents provision young, but only the female incubates eggs and broods young (*Woolfenden & Fitzpatrick, 1984*). Non-breeding helpers are seeking their first breeding opportunity. During this time, helpers reside at their natal or home territory and assist with its defense and vigilance against predators while also making repeated forays off-territory in search of potential breeding opportunities. Many routes to becoming a breeder exist, including pairing with a recently widowed breeder or unpaired dominant, establishing a territory in vacant habitat or, for males, inheriting all or a portion of their natal territory (*Woolfenden & Fitzpatrick, 1977*; *Woolfenden & Fitzpatrick, 1978*; *Woolfenden & Fitzpatrick, 1984*; *Woolfenden & Fitzpatrick, 1990*; *Breininger, 1999*; *Stith, 1999*). Dominant birds are those that are defending a territory but have yet to breed (defined as laying or siring an egg). They may be paired or unpaired. If unpaired, they may defend against many more potential usurpers than do established pairs and they also may foray, attempting to attract a mate.

If life history stage-specific tactics to maximize individual fitness influence social behavior, then social behavior should vary with life history stage. We draw on 50 years of study to predict how stage-specific behaviors would be reflected in the metrics calculated from the social network (Table 1). Breeders do not foray, which restricts the pool of individuals with which they can associate to family-group members, neighbors, and helpers foraying nearby. Helpers may foray frequently, and dominants may foray if unpaired, thus we predict that (1) breeders will have the fewest associations and helpers the most, (2) breeders will rarely connect otherwise unconnected individuals and helpers will do so frequently, (3) breeders will exhibit more "cliquish" behavior, associating with individuals that are themselves associated, and helpers will not and (4) breeders will be detected at the fewest unique locations and helpers at the most. Because there is one way to defend a territory, but many ways to obtain one, we predict that (5) breeders will exhibit the least within-stage variation and helpers the most.

## MATERIALS & METHODS

### In the field

We conducted this study on an individually-marked population of Florida scrub-jays at Archbold Biological Station in Highlands County, Florida (27°10′N, 81°21′W). As part of a long term and ongoing study, we band all scrub-jays with unique combinations of color bands and census the population monthly (see *Woolfenden & Fitzpatrick, 1984* for

**Table 1** *A priori* predictions about Florida scrub-jay *Aphelocoma coerulescens* social behavior based on life history stage.

| Metric | Breeders | Dominants | Helpers | Justification for predictions | Support for predictions |
|---|---|---|---|---|---|
| Within Stage Variance | low | high | high | • Little variation in how to maintain a territory and tend to young<br>• Dominants foray if unpaired, remain on territory if paired<br>• Helpers seek breeding opportunities and diverse routes to breeding increase variance in foray strategies | Mixed<br>• Differences in the amount of variation between life history stages was not consistent across years or metrics<br>• The variation in the number of unique points detected varied with life history stage in both years |
| Degree, number of individuals an individual was associated with | low | intermediate | high | • Breeders typically defend only against neighbors<br>• Dominant birds defend against usurpers and neighbors, may foray<br>• Helpers foray, increasing the pool of individuals with which they can interact | Mixed<br>• Breeders differed from dominants and helpers in 2017, but not 2018<br>• Dominants and helpers did not differ in either year |
| Betweenness, the importance of a focal individual in connecting others | low | intermediate | high | • Breeders interact with group members and neighbors, all directly connected<br>• Dominants interact with more foraying non-breeders during territory defense<br>• Helpers and unpaired dominants interact with individuals from non-adjacent territories during forays, connecting their group and neighboring groups to individuals from farther away | Mixed<br>• Breeders and helpers differed significantly in 2018<br>• Dominants and helpers differed significantly in 2018<br>• Breeders and dominants differed significantly in 2018 |
| Local clustering coefficient, proportion of an individual's associates which are themselves associated | high | intermediate | low | • Same rational as betweenness | Mixed<br>• Breeders differed from helpers in 2017, but not 2018<br>• Dominants differed from helpers in 2017, but not 2018 |
| Number of unique points detected | low | intermediate | high | • Breeders seen only at or near territory<br>• Dominant birds seen at or near territory and on occasional forays<br>• Helpers foray frequently and at longer distances | Yes<br>• Breeders significantly differed from dominants and helpers in both years |

a detailed description of our demographic study methods). The nesting season can begin as early as February and continues into June. Each year, we find all nests and map the boundaries of each territory.

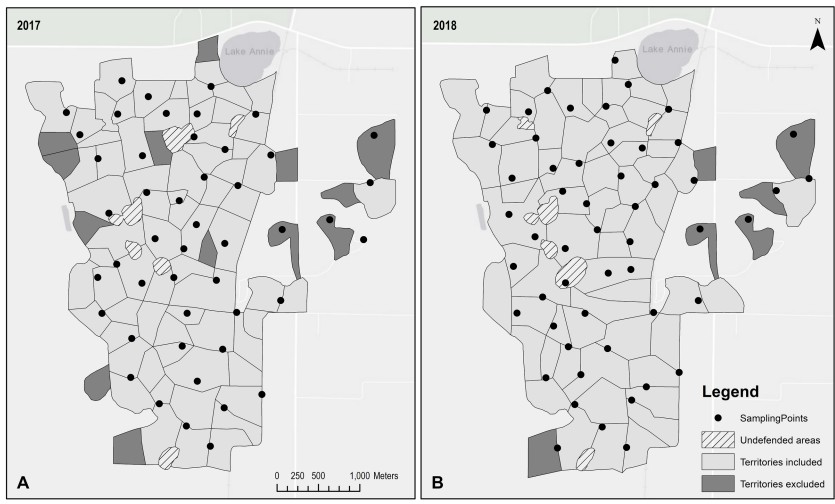

**Figure 1** **Location of Florida scrub-jay (*Aphelocoma coerulescens*) territories and aggregation sampling points at Archbold Biological Station in 2017 and 2018.** Boundaries shown here are as they were mapped in April of (A) 2017 and (B) 2018. Sampling points were non-randomly stratified with respect to territory boundaries as mapped the previous year and placed at least 200 m apart. Darkly shaded territories indicate that no individual from that territory was included in the statistical analyses, either because the individuals were detected too few times to be included in the network, or because the territory failed to meet the spatial criteria for inclusion in the analyses.

From February through April of 2016 and 2017, we surveyed for jays at points placed at least 200 m apart and non-randomly stratified across all territories (Fig. 1). Florida scrub-jays vigorously defend their territories, which average about 9 ha (*Woolfenden & Fitzpatrick, 1990*). While deep intrusion of one group's territory by another is relatively infrequent, mutual defense along shared boundaries is routine. Prospecting forays by non-breeding helpers tend to occur along these boundaries (*Woolfenden & Fitzpatrick, 1990*). Thus, we stratified sampling at the intersections of territory boundaries to ensure that (1) we captured interactions of birds foraging away from their home territory among themselves and with local birds and (2) all individual birds had an opportunity to be sampled, even those that did not foray. Because sampling began just prior to the onset of breeding and before we map territories, we based the placement of sampling points on the territory boundaries mapped in April of the previous year. Territory boundaries are relatively stable but do fluctuate annually and within a season (*Woolfenden, 1975*; *Woolfenden & Fitzpatrick, 1990*).

We sampled these points twice a week in 2017 and 2018 using playback of territorial calls. Calls were sourced from recordings made in the same metapopulation as our study population (*Coulon et al., 2008*), but are several decades old. Thus, these calls are in the local dialect, but not from individuals known to any jays in our study. We randomized the order in which we visited points using a random number generator. We played territorial calls on a portable speaker for a total of 2 min, 30 s in each cardinal direction, with 30-second breaks after each. Then we waited an additional minute for birds to respond, so that each

visit lasted a minimum of 5 min. We recorded the identities of all birds present at survey points using Survey123 (Esri, Redlands, CA, USA).

Use of playback is appropriate for this species for two reasons: (1) it mimics what occurs when there is a disturbance in the existing social structure, such as the death of a breeder, which are the types of opportunities foraying helpers are searching for and (2) breeders, dominants, and helpers all defend their territorial boundaries. In order for individuals to hear and respond to the playbacks, they must already be in the vicinity of the sampling point. Territories are large and scrub habitat is generally low and open, and we do not observe jays flying in from far away in response to the playbacks, which supports the idea that the individuals responding to playback are nearby. Florida scrub-jays are vigilant, territorial, and social, making them likely to detect jays along or within their territory boundary and interact with them. Therefore, we assume that individuals detected at a sampling point at the same time are associated. We have no reason to suspect that the associations we observe during sampling would not exist in the absence of the playbacks.

To maximize the number of edges recorded (*Davis, Crofoot & Farine, 2018*) between jays from non-adjacent territories, we also recorded opportunistic observations of aggregations of jays when they contained individuals from non-adjacent territories. These aggregations are ephemeral, and it is difficult to predict when or where they will occur. We did not record opportunistic observations of members of the same territory or neighboring territories because these edges are easily captured by the point sampling, which occurs along relatively static and vigorously defended territory boundaries. Because ignoring opportunistic observations of individuals from the same or neighboring territories interacting underestimates the strength of connections between family members and neighbors, we used binary, and not weighted, degree.

All research was conducted under the required permits from the United States Geological Survey Bird Banding Lab (07732) and the United States Fish and Wildlife Service (TE824723-9) issued to RB.

## Constructing the social networks

We included all individuals detected three or more times in group-by-individual matrices for each year. We chose this cutoff because it reduced the proportion of dyads below the recommended threshold simple ratio index denominator of 20 (*Farine & Strandburg-Peshkin, 2015*; *Davis, Crofoot & Farine, 2018*) without excluding too large a portion of the female breeders. We used R package asnipe to build a network for each year, using the simple ratio index to correct for detection probabilities (*Whitehead, 2008*; *Farine & Whitehead, 2015*; *Hoppitt & Farine, 2018*). We chose three metrics that we thought would best capture the differences in behavior between helpers, who make frequent off-territory forays, and breeders, who do not (Table 2): (1) binary degree, the number of individuals an individual was associated with; (2) vertex betweenness centrality (hereafter betweenness), the importance of a focal individual in connecting other birds or connected groups, i.e., the number of times a focal individual lies on shortest paths between two other individuals; and (3) local clustering coefficient, the proportion of the focal individual's associates which are themselves associated (*Croft, James & Krause, 2008*; *Beveridge & Shan, 2016*). We
**Table 2 Individuals in the population (counted April of the study year), network, and analyses.** Individuals that were included in the network and survived through the sampling period, but were not included in the analyses were from territories that failed to meet the spatial criteria for inclusion.

| Year | 2017 | | | | | | 2018 | | | | | |
|---|---|---|---|---|---|---|---|---|---|---|---|---|
| Stage | Breeder | | Dominant | | Helper | | Breeder | | Dominant | | Helper | |
| Sex | F | M | F | M | F | M | F | M | F | M | F | M |
| Study population | 69 | 73 | 10 | 7 | 22 | 28 | 57 | 58 | 11 | 11 | 28 | 29 |
| Included in network | 56 | 64 | 9 | 7 | 24 | 27 | 62 | 60 | 9 | 10 | 32 | 28 |
| Analyzed | 49 | 56 | 9 | 7 | 20 | 26 | 48 | 50 | 9 | 10 | 29 | 26 |
| n | | | 167 | | | | | | 172 | | | |
| % dyads SRI denominator <20 | | | 30% | | | | | | 6% | | | |
| (Denominator mean ± SE) | | | (30.16 ± 0.12) | | | | | | (42.14 ± 0.11) | | | |

calculated these metrics using R package igraph (*Csardi & Nepusz, 2006*), specifying the weights argument for betweenness as the inverse of the edge weights (*Silk, 2018*). We also counted the number of unique sampling points at which each individual was detected.

## Statistical analyses

We adopted spatial criteria for inclusion in our statistical analyses because yearly changes to territory boundaries could result in uneven sampling coverage and because territories along the periphery have fewer neighboring territories and tended not to have helpers. We calculated the distance between the territory edge and the nearest sampling point and number of immediately adjacent territories for all territories as they were mapped in the year they were sampled using the Generate Near Table and Polygon Neighbors tools in ArcMap (Esri, Redlands, CA, USA).

Individuals from territories that did not have a sampling point within 100 m of their territory boundary and those from territories with one or fewer immediately adjacent neighboring territories were excluded from the statistical analyses. Then we used ANOVA to confirm that that individuals in different life history stages did not differ in the distance between the territory on which they reside and the nearest sampling point (2017: $F_{2,164} = 1.04$, $p = 0.36$; 2018: $F_{2,169} = 2.46$, $p = 0.09$) nor the number of adjacent territories (2017: $F_{2,164} = 1.71$, $p = 0.18$; 2018: $F_{2,169} = 3.23$, $p = 0.04$). In 2018, dominant birds resided on territories significantly farther from the nearest sampling point than helpers ($p = 0.01$) and breeders ($p = 0.05$), but the observed differences were less than 15 m. In 2018, helpers resided on territories with nearly 1 more adjacent territory than breeders ($p = 0.05$) and dominants ($p = 0.05$), likely an artifact of territories in the center of our study area tending to be more productive.

We also estimated the robustness of our social networks for each year. For each dyad in each year, we calculated the denominator of the simple ratio index (the number of observations in which either individual was detected). The proportion of dyads at or above the minimum simple ratio index threshold of 20 (*Farine & Strandburg-Peshkin, 2015*; *Davis, Crofoot & Farine, 2018*), increased from 2017 to 2018 (Table 2).

We constructed a set of linear models to explain each social network metric in each year by sex, life history stage, their interaction, the number of territories adjacent to their

territory, and the distance between their territory and the nearest sampling point. We calculated the proportion of variance explained by each parameter, $\eta2$, as the sum of squares divided by the residual sum of squares.

To determine if the social network metrics differed between life history stages, we used the same set of models and calculated between group differences using the TukeyHSD() command in R (*R Core Team, 2018*). To determine if variances differed among life history stages we used Brown-Forsythe tests. We compared the observed differences and F statistics to those calculated from one thousand data-stream permutations. We computed the data-stream permutations using the network_permutation() command in R package asnipe, which allowed us to control for the number of times an individual was observed as well as the location of those observations (*Farine & Whitehead, 2015*; *Farine, 2017*). Then we calculated one-tailed p-values to test for significant differences.

## RESULTS

In April of 2017, 209 individuals were in the population (Table 2). In 2017, we made 1104 sampling observations of 215 individuals. Of these individuals, 191 were detected three or more times and included in the 2017 network. We analyzed data for 167 of the 191 included in the network (Table 2). We excluded six individuals that did not survive to the end of the sampling period, nine breeders and one helper from territories that did not meet the maximum distance criterion and six breeders and two helpers from territories that did not meet the number of adjacent territories criterion. In April of 2018, 194 individuals were in the population. The network was based on 1324 observations of 206 individuals. From our analysis we excluded 12 individuals who did not survive the sampling period, three who were moving on and off our study tract, and 17 breeders and 2 helpers from territories that did not meet the spatial inclusion criteria.

The results conformed to some, but not all of our predictions (Table 1). Life history stage explained a high proportion of the variance observed in all metrics in all years (Table 3). Breeders tended to have fewer connections, lower betweenness, higher clustering coefficients, and visited fewer unique points than helpers, and dominants were intermediate (Fig. 2, Table 4). However, breeders did not have consistently less variation in their social network metrics than helpers (Fig. 2, Table 5).

Dominants behaved as predicted, typically having metric scores between those of breeders and helpers (Fig. 2). However, these differences were not consistent across metrics or years (Table 4). Dominants were detected at more unique points than breeders in all years and had higher degree than breeders in 2017. Dominants had lower betweenness than helpers in 2018 and higher clustering coefficient in 2017.

Helpers differed significantly from breeders in all metrics except betweenness in 2017 (Table 4). Helpers had higher degree and betweenness, lower clustering coefficients, and were detected at more unique points than breeders (Fig. 2). These differences were statistically significant for all metrics except betweenness in 2017 and in 2018, only the differences in betweenness and the number of unique points detected were statistically significant. Female helpers tended to have higher degree and betweenness, lower clustering
**Table 3  Life history stage explains the highest proportion of explained variance, $\eta^2$, for all measured variables in both years.**

| Metric | Factor | 2017 | | | | 2018 | | | |
|---|---|---|---|---|---|---|---|---|---|
| | | SS | df | F | $\eta^2$ | SS | df | F | $\eta^2$ |
| Degree | Distance to Nearest Point | 280.00 | 1.00 | 1.19 | 0.01 | 643.00 | 1.00 | 3.53 | 0.02 |
| | Number of Adjacent Territories | 9019.00 | 1.00 | 38.21 | 0.24 | 11695.50 | 1.00 | 64.17 | 0.39 |
| | Life History Stage | 14190.00 | 2.00 | 30.06 | 0.38 | 13759.90 | 2.00 | 37.75 | 0.46 |
| | Sex | 15.00 | 1.00 | 0.06 | 0.00 | 166.40 | 1.00 | 0.91 | 0.01 |
| | Life History Stage *Sex | 868.00 | 2.00 | 1.84 | 0.02 | 203.50 | 2.00 | 0.56 | 0.01 |
| | Residuals | 37534.00 | 159.00 | | | 29890.20 | 164.00 | | |
| Betweenness | Distance to Nearest Point | 0.0000 | 1.00 | 0.00 | 0.00 | 0.0002 | 1.00 | 0.43 | 0.00 |
| | Number of Adjacent Territories | 0.0016 | 1.00 | 2.06 | 0.01 | 0.0002 | 1.00 | 0.49 | 0.00 |
| | Life History Stage | 0.0100 | 2.00 | 6.58 | 0.08 | 0.0110 | 2.00 | 12.60 | 0.15 |
| | Sex | 0.0004 | 1.00 | 0.54 | 0.00 | 0.0002 | 1.00 | 0.54 | 0.00 |
| | Life History Stage *Sex | 0.0028 | 2.00 | 1.83 | 0.02 | 0.0001 | 2.00 | 0.16 | 0.00 |
| | Residuals | 0.1207 | 159.00 | | | 0.0714 | 164.00 | | |
| Clustering Coefficient | Distance to Nearest Point | 0.000 | 1.00 | 0.00 | 0.00 | 0.026 | 1.00 | 1.46 | 0.01 |
| | Number of Adjacent Territories | 0.541 | 1.00 | 26.75 | 0.17 | 0.207 | 1.00 | 11.79 | 0.07 |
| | Life History Stage | 1.507 | 2.00 | 37.24 | 0.48 | 0.485 | 2.00 | 13.80 | 0.17 |
| | Sex | 0.004 | 1.00 | 0.22 | 0.00 | 0.003 | 1.00 | 0.15 | 0.00 |
| | Life History Stage *Sex | 0.021 | 2.00 | 0.52 | 0.01 | 0.007 | 2.00 | 0.19 | 0.00 |
| | Residuals | 3.137 | 155.00 | | | 2.880 | 164.00 | | |
| Unique Points Detected | Distance to Nearest Point | 23.29 | 1.00 | 4.68 | 0.03 | 8.31 | 1.00 | 1.75 | 0.01 |
| | Number of Adjacent Territories | 174.86 | 1.00 | 35.16 | 0.23 | 194.03 | 1.00 | 40.90 | 0.25 |
| | Life History Stage | 362.22 | 2.00 | 36.42 | 0.48 | 373.13 | 2.00 | 39.32 | 0.48 |
| | Sex | 1.41 | 1.00 | 0.28 | 0.00 | 8.46 | 1.00 | 1.78 | 0.01 |
| | Life History Stage *Sex | 18.61 | 2.00 | 1.87 | 0.02 | 13.37 | 2.00 | 1.41 | 0.02 |
| | Residuals | 760.94 | 153.00 | | | 778.13 | 164.00 | | |

coefficients, and to be detected at more unique points than males (Fig. 2). However, the magnitude of the sex differences in degree and clustering coefficient only reached statistical significance in 2017 (Table 4). The differences in betweenness and the number of unique points female and male helpers were detected was not significant in any year.

# DISCUSSION

We examined how social network position is shaped by life history stage in a cooperatively breeding bird. Our results show that life history stage explains much of the observed variation in social network position during the pre and early breeding season. The social network metrics we calculated were generally consistent with our predictions based on nearly 50 years of observations of Florida scrub-jay natural history (*Woolfenden, 1975*; *Stallcup & Woolfenden, 1978*; *Woolfenden & Fitzpatrick, 1984*; *Woolfenden & Fitzpatrick, 1990*; *Stith, 1999*). However, breeder behavior was more variable than expected.

Our sampling period began approximately one month before the onset of nesting and ended shortly after most pairs had active nests with eggs or young (*Woolfenden & Fitzpatrick, 1984*). During the breeding season, breeders must defend their territory and
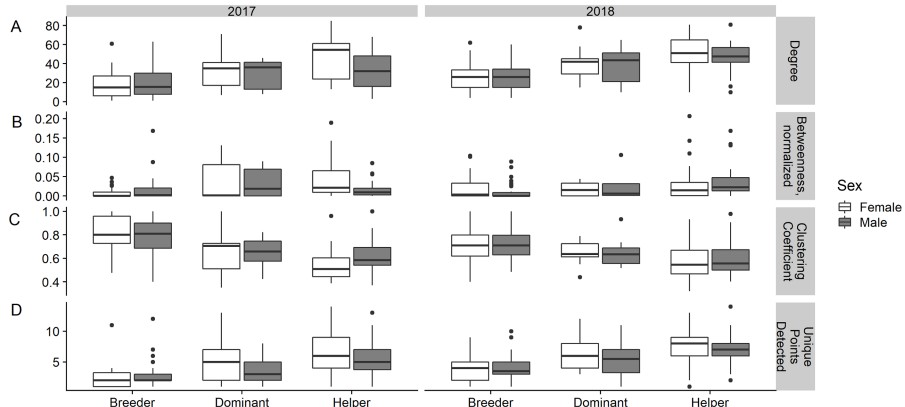

**Figure 2** Boxplots showing degree, betweenness, clustering coefficient, and number of unique points visited for female and male Florida scrub-jays (*Aphelocoma coerulescens*) by life history stage. (A) Breeders had significantly fewer connections (lower degree) than helpers in 2017. (B) Breeders connected otherwise unconnected individuals (lower betweenness) significantly less than helpers in both years. (C) Breeders exhibited significantly more "cliquish" behavior (higher clustering coefficient) than helpers in 2017. (D) Breeders were detected at significantly fewer unique points than helpers in 2017 and 2018.

tend their nest (*Stallcup & Woolfenden, 1978*). Thus, breeders interacted primarily with their neighbors and group members, which is reflected in all of the variables we measured. Breeders were detected at relatively few unique points because they remained on or near their territory. This restricted the pool of individuals with which they associated to primarily group-members and neighbors, and occasionally helpers foraying nearby, leading to a paucity of connections. Group members and neighbors are already directly connected to one another, thus breeders have little opportunity to connect otherwise unconnected individuals, resulting in their low betweenness and high clustering coefficient.

During our sampling period, helpers frequently forayed off their natal territory, occasionally forming aggregations of unrelated helpers (*Woolfenden & Fitzpatrick, 1990*). Forays provide an opportunity to interact with individuals from non-adjacent territories. These forays explain why helpers are detected at more unique points, as well as their higher degree and betweenness, and lower clustering coefficient. By interacting with jays from non-adjacent territories, helpers indirectly connected their group members and neighbors to the families and neighbors of individuals with which they associated on forays. Female Florida scrub-jays disperse earlier and farther than males (*Woolfenden & Fitzpatrick, 1978*; *Woolfenden & Fitzpatrick, 1984*; *Woolfenden & Fitzpatrick, 1990*), thus we expected that female helpers would have higher degree and betweenness, lower clustering coefficients, and visit more unique points. However, we only detected significant differences between male and female helpers in 2017 and only in degree and clustering coefficient. Despite females being the dispersing sex, they were not detected at more unique points than males in either year.

Like breeders, dominants must defend their territory and, like helpers, they may be searching for a mate, either by foraying or waiting for foraying helpers to visit their territories. Dominants' metrics were intermediate between those of helpers and breeders,

**Table 4   Observed and randomized differences in social network metrics ± 95% CI between life history stages in Florida scrub-jay (*Aphelocoma coerulescens*).** Observed differences were calculated using Tukeys honestly significant differences. Randomized difference and *p*-values for metrics calculated from the social network were estimated using 1,000 data stream permutations. *P*-values < 0.05 are in bold.

| Comparison | Metric | Prediction | Year | Observed difference ± 95% confidence level | Mean randomized difference ± 95% confidence level, 1000 permutations | p |
|---|---|---|---|---|---|---|
| Breeders vs. Dominants | Degree | lower | 2017 | −17.12 ± 9.76 | −16.18 ± 0.04 | **<0.001** |
| | | | 2018 | −14.19 ± 8.00 | −13.76 ± 0.04 | 0.23 |
| | Betweenness | lower | 2017 | −0.029 ± 0.019 | −0.010 ± 0.0007 | 0.09 |
| | | | 2018 | −0.004 ± 0.020 | −0.002 ± 0.0001 | **0.04** |
| | Clustering Coefficient | higher | 2017 | 0.173 ± 0.091 | 0.161 ± 0.0006 | 0.10 |
| | | | 2018 | 0.068 ± 0.079 | 0.081 ± 0.0002 | 0.99 |
| | Unique Points | fewer | 2017 | −2.66 ± 1.42 | | **<0.001** |
| | | | 2018 | −2.43 ± 1.29 | | **<0.001** |
| Dominants vs. Helpers | Degree | lower | 2017 | −2.56 ± 10.55 | −2.10 ± 0.03 | 0.14 |
| | | | 2018 | −4.70 ± 8.50 | −5.44 ± 0.04 | 0.79 |
| | Betweenness | lower | 2017 | 0.010 ± 0.020 | −0.005 ± 0.0007 | 0.84 |
| | | | 2018 | −0.017 ± 0.021 | −0.004 ± 0.0002 | **<0.001** |
| | Clustering Coefficient | higher | 2017 | 0.032 ± 0.098 | 0.018 ± 0.0003 | **<0.001** |
| | | | 2018 | 0.046 ± 0.083 | 0.039 ± 0.0003 | 0.12 |
| | Unique Points | fewer | 2017 | −0.60 ± 1.55 | | 0.63 |
| | | | 2018 | −0.66 ± 1.37 | | 0.49 |
| Breeders vs. Helpers | Degree | lower | 2017 | −19.68 ± 6.43 | −18.28 ± 0.02 | **<0.001** |
| | | | 2018 | −18.90 ± 5.38 | −19.20 ± 0.03 | 0.80 |
| | Betweenness | lower | 2017 | −0.018 ± 0.012 | −0.016 ± 0.0002 | 0.18 |
| | | | 2018 | −0.021 ± 0.013 | −0.007 ± 0.0002 | **<0.001** |
| | Clustering Coefficient | higher | 2017 | 0.206 ± 0.060 | 0.179 ± 0.0004 | **<0.001** |
| | | | 2018 | 0.115 ± 0.053 | 0.121 ± 0.0002 | 0.98 |
| | Unique Points | fewer | 2017 | −3.26 ± 0.96 | | **<0.001** |
| | | | 2018 | −3.09 ± 0.87 | | **<0.001** |
| ♀ vs. ♂ Helpers | Degree | higher | 2017 | 6.54 ± 13.18 | 1.87 ± 0.03 | **<0.001** |
| | | | 2018 | 4.26 ± 10.51 | 5.90 ± 0.03 | 0.99 |
| | Betweenness | higher | 2017 | 0.022 ± 0.025 | 0.025 ± 0.0002 | 0.75 |
| | | | 2018 | 0.006 ± 0.026 | −0.001 ± 0.0001 | 0.99 |
| | Clustering coefficient | lower | 2017 | −0.025 ± 0.122 | 0.026 ± 0.0002 | **<0.001** |
| | | | 2018 | −0.024 ± 0.103 | 0.018 ± 0.00054 | 0.20 |
| | Unique points | more | 2017 | 0.89 ± 1.98 | | 0.79 |
| | | | 2018 | 1.08 ± 1.70 | | 0.45 |

as predicted by the natural history observations. Dominants were detected at more points than breeders in all years and had higher degree in 2017. Dominants had significantly lower betweenness than helpers in 2018, and higher clustering coefficient in 2017, but did not

**Table 5  Brown-Forsythe tests for homogeneity of variance across life history stages.** The variance of degree, betweenness, clustering coefficient, and the number of unique points at which an individual was detected are not equal among life history stages in some years.

| Metric | Year | Observed F Statistic for homogeneity | Mean randomized F ± 95% Confidence level, 1000 permutations | p |
|---|---|---|---|---|
| Degree | 2017 | 10.38 | 9.67 ± 0.05 | 0.19 |
| | 2018 | 1.47 | 0.61 ± 0.01 | **<0.001** |
| Betweenness | 2017 | 9.66 | 5.62 ± 0.11 | **0.02** |
| | 2018 | 4.99 | 4.89 ± 0.15 | 0.44 |
| Clustering coefficient | 2017 | 0.59 | 0.67 ± 0.02 | 0.58 |
| | 2018 | 0.82 | 2.24 ± 0.03 | 0.99 |
| Unique Points Detected | 2017 | 10.89 | | **<0.001** |
| | 2018 | 3.50 | | **0.03** |

differ in the number of unique points at which they were detected in any year. For helpers, we inferred that their high betweenness and low clustering coefficients were driven by their foraying, which is reflected in the high number of unique points at which they were detected. However, dominants and helpers did not differ in the number of unique points at which they were detected but tended to differ in betweenness and clustering coefficient. This suggests that the identity of the points at which an individual was detected, and not only the number of unique points, may drive some of the variation in betweenness and clustering coefficient. Dominants' low betweenness and high clustering coefficients relative to the number of unique points they were detected could also be explained by increased territorial intrusion from neighboring family groups. Territorial intrusions will mainly be from groups with which the dominant bird shares a territorial border. These intrusions may draw individuals residing on opposite sides of a dominant's territory together which could increase dominants' clustering coefficients and decrease betweenness.

The quantitative social network metrics we calculated were generally consistent with the qualitative observations previously published. However, the differences we predicted were not statistically significant in all years and we observed more variation among breeders than predicted. Yearly differences in demography, breeding opportunities, or both, driven by environmental variation, may explain why we failed to detect significant differences between breeders and helpers in 2018 that were evident in 2017. Between 2017 and 2018, the number of scrub-jay family groups declined by 10, resulting in fewer breeders in our population. Yearly changes in social behavior could be a passive reflection of, rather than an active response to, changes in the social landscape. When individuals are removed or added to the network the existence and arrangement of relationships in the network change (*Shizuka & Johnson, 2019*) and changes to group composition can reduce the repeatability of social network metrics (*Plaza et al., 2019*). Additionally, individual personality affects behavior (*Aplin et al., 2013*) and the sets of individuals in any given life history stage are different every year. Therefore, some amount of annual variation may be attributable to demographic changes rather than behavioral plasticity.

Alternatively, the yearly variation we observed may be due to behavioral responses to environmental conditions or the probability of breeding successfully. During 2018, the scrub-jays experienced unusually low reproductive success. Nest success was much lower (32% versus 55% in 2017) and many pairs did not attempt to breed at all; only 56 of the 75 groups produced nests with contents. Once a breeding pair lays their first egg, their behavior changes dramatically: the breeding female incubates or broods, the breeding male provisions both female and nestlings. Prior to laying, the behavior of breeding birds may be much more like dominants. Thus, we reran the analysis for 2018 with the breeders that did not lay eggs categorized as dominants, but as in the previous analysis we failed to detect significant differences in degree or clustering coefficient between breeders that laid eggs and helpers. However, in 2018 the onset of breeding also was delayed. On average, pairs began laying 20 days later in 2018 than in 2017. Because our observational sampling period ends near the end of the first third of the breeding season, we had many fewer sampling days where even breeders that eventually laid eggs, had nests with contents, and thus may have been behaving more like dominants. It does suggest that not only life history stage, but individual life-histories and the environment can influence social behavior.

Alternatively, the similarities we observed between breeders and helpers in 2018 could be explained by helpers reducing their foray behavior. If helpers perceived that their chances of breeding in 2018 were low, they may have reduced foraying behavior, choosing to further delay dispersal until a year with better prospects. However, we see no evidence of reduced helper foray behavior in 2018 and their social metrics were similar between 2017 and 2018. Therefore, we conclude that extra-territory socialization is important regardless of life history stage, but may be influenced by a variety of factors that might alter the costs and benefits of conducting off-territory forays. When not actively tending nests with eggs or young, breeders may allocate more time to interacting with individuals from other groups.

Current thinking frames the social behavior of Florida scrub-jay helpers as part of a strategy to maximize their probability of obtaining a breeder position. However when breeders do not have an active nest they socialize with a similar number of individuals, exhibit similar amounts of cliquishness, and visit a similar number of unique points as individuals who are seeking a territory and mate. This suggests that extra-territorial social behavior during the pre and early breeding season has adaptive value beyond establishing a territory and finding a mate. Social interactions between neighbors may reduce the overall costs of territory defense, either by establishing boundaries prior to the increased energetic demands associated with breeding, or by establishing coalitions to repel potential usurpers (*Temeles, 1994*; *Christensen & Radford, 2018*). Increased sociality also may serve to reduce predation risk or time spent in vigilance (*Groenewoud et al., 2016*; *Mady & Blumstein, 2017*; *Van der Marel, López-Darias & Waterman, 2019*).

The physical environment and spatial distribution and availability of resources influence social behavior, network structure and the transmission of information (*Slobodchikoff, 1988*; *Foster et al., 2012*; *Webster et al., 2013*; *Leu et al., 2016*; *He, Maldonado-Chaparro & Farine, 2019*). Each year, we observe aggregation hotspots, where helpers tend to aggregate frequently and in large numbers. These spots change yearly, but little is known about what drives this variation. Helpers may be cueing in on habitat quality, the availability of

undefended habitat, or potential mates. Alternatively, they may be relying on information transmitted through the social network about the location of these aggregations. Regardless of what determines hotspot locations, helpers' attraction to them has the potential to affect the network metrics of individuals in the vicinity. Additionally, helpers contribute to nest and territory defense and vigilance against predators and their presence may allow breeders with helpers to engage in more social behavior, especially with non-group members, than breeders without helpers. Thus, individuals in one life history stage may influence the network position of individuals in other stages, even if they are not exploiting the same resource.

Ecological needs and selection pressures change with life history stage. Therefore, life history stage can have profound impacts on an individual's behavior and social network position. Social behavior is adaptive (*Alexander, 1974*; *Silk, Alberts & Altmann, 2003*), and current social behavior can affect future reproduction (*McDonald, 2007*). We demonstrate that life history stage can explain much of the observed variation in individual social network position and that breeding may constrain social behavior. Because many social species have stage-structured populations, it is important to consider both the effect of life history stage and breeding status on individual position within the social network.

## CONCLUSIONS

We hypothesized that behavior within a social network would reflect life history stage in the cooperatively breeding Florida scrub-jay. We found that social behavior varies with life history stage and between years. Research into whether individuals employ different socialization strategies depending on the types and location of breeding opportunities is ongoing. Our results demonstrate the power of life history stage to explain variation in social behavior and suggest that social relationships between members of different groups may be more important than previously realized.

## ACKNOWLEDGEMENTS

The authors thank S. Carrera-Lozano, L. Clark, M. Fuirst, A. Gonzalez, J. Greer, M. Heather, P. Hopkins, H. Kenny, S. Prussing, Y. Suh, R. Wadleigh and R. Windsor for data collection, V. Sclater for assistance with Esri products, and all of the support staff at Archbold Biological Station.

### Funding

This work was supported by Archbold Biological Station. The funders had no role in study design, data collection and analysis, decision to publish, or preparation of the manuscript.

### Grant Disclosures

The following grant information was disclosed by the authors:
Archbold Biological Station.

## Competing Interests

The authors declare there are no competing interests.

## Author Contributions

- Angela Tringali conceived and designed the experiments, analyzed the data, prepared figures and/or tables, authored or reviewed drafts of the paper, and approved the final draft.
- David L. Sherer conceived and designed the experiments, performed the experiments, analyzed the data, authored or reviewed drafts of the paper, and approved the final draft.
- Jillian Cosgrove conceived and designed the experiments, performed the experiments, authored or reviewed drafts of the paper, and approved the final draft.
- Reed Bowman conceived and designed the experiments, authored or reviewed drafts of the paper, and approved the final draft.

## Animal Ethics

The following information was supplied relating to ethical approvals (i.e., approving body and any reference numbers):

All research is done under permits from the United States Geological Survey (bird capture and banding) and the United States Fish and Wildlife Service (research on federally listed species) (USGS Banding Permit 07732; USFWS Threatened and Endangered Species Permit TE824723-9).

## Data Availability

Data and code used for analysis is available at Figshare: Tringali, Angela (2019): Life history stage explains social behavior. figshare. Dataset. https://doi.org/10.6084/m9.figshare.8097488.v1.

## Supplemental Information

Supplemental information for this article can be found online at http://dx.doi.org/10.7717/peerj.8302#supplemental-information.

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
