# Peer review of "Life history stage explains behavior in a social network before and during the early breeding season in a cooperatively breeding bird"

_PeerJ, doi:10.7717/peerj.8302_

## Round 0.1 · original submission · Major Revisions

Thank you for submission to PeerJ. I have been fortunate to receive reviews from two experts in your field. Both had major concerns with your article, relating to the aims of your study and to the manner with which you carried out your analyses.

That said, I would like to give you an opportunity to respond to the reviewers' concerns. This will require carefully responding to all of the reviewers' comments, and likely making major changes to the paper and analyses. I cannot decide the eventual publishability of this work without seeing how these concerns are addressed.

I look forward to receiving a revised version of your article.

·

Basic reporting

See review below

Experimental design

See review below

Validity of the findings

See review below

Additional comments

The authors present an interesting study looking at differences in social network position of individuals across different life stages. This is a neat idea, and the study is reasonably well presented. But, I’m afraid that I think there is a potentially major confound that hasn’t been sufficiently accounted for, which may have some impact on the results. In addition, it isn’t clear that the networks are sufficiently robust to conduct these analyses; at least not with the current hypothesis-testing approach.

The potentially major issue with the analysis as it stands is that the majority of the results could be explained by one factor – the number of observations of each individual. The more often an individual is observed, the more opportunity there are to observe connections – thus individuals with more observations typically have a higher degree (# obs also affects other network metrics too). It often takes very many observations to remove this effect. In our great tit networks the threshold was c. 100 observations per individual(!). As the authors included their data with their submission, I had a quick look at the potential for this effect. For 2016, I found that probably all of their patterns are explained by the number of observations. The correlation between #obs with degree was 0.81, with betweenness was 0.56, and with clustering coefficient was -0.67. However – don’t despair! The very simple solution to this is to switch null models to one that explicitly and correctly controls for number of observations (please don’t add #obs into a linear model because the relationship is rarely linear). Doing pre-network permutations would allow the authors to do this – in fact the code to do this is hardly more complicated than the existing approach they use (it’s just one command in the asnipe package). At the same time, the authors could also control for space (by randomising within sampling point). Should the authors have any questions – I always reply promptly to any questions regarding the application of my R packages. There is also useful examples of code in the appendices of the Farine & Whitehead 2015 paper which is cited in this manuscript.

Building a little bit on the previous point, another issue that is present with the current work is whether the networks are robust. The authors cut off their minimum data at 3 observations. In Farine & Strandburg-Peshkin 2015 (RSOS) and Davis et al. 2018 (Animal Behaviour), we first provide guidance on how to estimate robustness, and also suggest that a good rule of thumb is that dyads should have the denominator of the simple ratio index sum up to 20. It could be that this is the case at present, but it’s unclear from the existing text.

Moving away from the methods a bit and to the broader perspective – it wasn’t clear to me what a network really means in a territorial system such as this. I understand that one might study within-group networks, but beyond this I can’t really pin much meaningful biology on it. For example, what does it mean to have a higher degree? Birds in larger ‘family’ units will have a higher degree, and maybe those with more nearby neighbours. But in the broader context, what does a population-level network capture? I don’t have an answer to this, but feel that the authors should provide one. On that note, it also seems a bit internally inconsistent to state that data were collected after conducting playbacks (L141) and then going on to state that this did not create artificial associations (L146).

I also had a question in terms of field data collection, related to the point immediately above. On L154, the authors state that ‘We assumed that all birds detected together at a point were associated’. This seems fine. But, in the next sentence, they state that ‘We also recorded opportunistic observations of aggregations of jays from non-adjacent territories.’ First I don’t understand what this means – but if my interpretation is correct (which is that you distinguished between groups that were from the same territory and those that were from different territories), I worry that a priori information about the social structure of the population is creeping into the field data being collected. This should be avoided at all costs. (Note-I work on a system where birds live in stable groups, and it’s an ongoing battle to stop students from incorporating their knowledge of system into their observations).

From my reading of the manuscript, I’d say that the authors will have no troubles addressing the comments outlined above. However, it may be that their results won’t be so clear. I hope that this won’t prevent them from publishing the manuscript irrespective of whether the current results stand or not—but I do encourage the authors from publishing a manuscript based on the most robust analysis they can achieve.

Damien Farine
(I sign all of my reviews)

Minor comments:
L30: Here and throughout the opening section of the introduction the citations seem very old. The first 7 citations are over 10 years old. I’d argue, for example, that the first citation does not show evidence that networks are a powerful way of quantifying social behaviour. The work done since then does.

L32: In Farine 2015 (Animal Behaviour) I deal with this question of interactions vs. observations more explicitly than the current citations.

L46: In the broader context of social associations, Alberts 2019 JAE should be cited here.

L56: The first sentence here reads really awkwardly.

L60: The sentence here doesn’t make much sense. You could have an accurate representation of the real world that is difficult to predict. For example, we can accurately measure earthquakes, but we cannot predict them.

L63: Why are you evaluating social networks? This ‘aim’ seems to pop up a few times in the introduction and elsewhere but it isn’t well justified or explained.

L82: At the end of the sentence here it feels that there should be a point made, but it isn’t made.

L100: Why is this a hypothesis? Where is the justification for calling it a hypothesis?

Table 1: I suggest adding a column that summarises the support (or not) for each prediction (see Brandl et al. 2019 Proceedings B for an example).

L141: The use of the playbacks brings into question whether what is being measured here area really social networks. They are something – but not really social networks in the context of how the introduction is framed. At least some more biological justification is required here.

L146: How are they not artificial? Would you expect that the associations that you have observed would have happened if you hadn’t performed the playbacks? How would you know this?

L161: Why choose 3? What is the justification behind this choice?

L164: It would be useful to give some more justification for why these three metrics were chosen. What are some biological insights that have been gained from them?

L189: I don’t understand the use of ‘distance to nearest sampling point’. If you sample individuals only at sampling points, how do you estimate this? Is it a property of the territory they are living on?

L195: I totally failed to understand what the difference is between the two sets of models. I re-read this section at least 10 times, and still couldn’t come away with a difference. The results seem to contain only those for one set of models (Table 3). The only difference between these two sets of methods is that the first says ‘each social network variable’ and the second says ‘social network metrics’ – but the first fits sex, etc. suggesting an individual-level analysis (and therefore applying to social network metrics).

L197: Why did you sqrt the degree? Was this done for the previous model as well? Needs a justification.

L199: Node permutations do not control for number of observations or space. (also were the randomised data only swapped within year?)

Reviewer 2 ·

Basic reporting

No comment

Experimental design

no comment

Validity of the findings

no comment

Additional comments

The aim of this study was to determine the extent to which life history stages of Florida scrub-jay, a cooperatively breeding bird, corresponds to position in a social network resulting from life history stage-specific social behaviour. The authors indeed report substantial differences in network connectivity observed, for instance, between helpers and breeders. The authors conclude that their network-derived results are in line with a priori expectations based on previous longitudinal studies on the subject. While the topic of the study is interesting, the described data collection procedures are admirable, I have one major comment. I struggle to understand the main aim of this study or, specifically, the rationale behind the study design except the apparent novelty resulting from the application of the SNA method. As currently written, it is rather unclear whether the application of SNA in this study adds to what have been previously known regarding the interplay between social behaviour and life history stages in cooperatively breeding species. Can the methods used in this ms shed some new light on previous studies (e.g., that used non-SNA methods) regarding the social behaviour of cooperatively breeding species? Which aspects of SNA results, if any, lead to different conclusions compared to what would be expected by applying methods based solely on dyadic interactions? Some of the results presented here provide a hint to such questions. For example, discussing the variation in helper network connectivity in relation to transitioning to breeders I find very interesting indeed (although I have some serious doubts regarding the chosen methods as well as the interpretation of the results-see below in specific comments) and I think that this aspect of the ms should be elaborated. Therefore, I would suggest to re-frame the focus of this ms and put more emphasis on the network position variation within a specific life history stage (e.g., helpers, and how this variation relates to a potential reproductive success) rather than merely making a case for demonstrating that SNA is capable of replicating what other methods do. Such refocus would also allow for making much more specific and biologically relevant hypotheses compared to what is currently presented. I have also several specific comments regarding both the theoretical side of the paper and statistical analysis:
. Line 30 why is SNA ‘powerful’, how does it relate to social behaviour derived from dyadic observations only? It has to be explained for a general audience as not everyone is a SNA expert
. Line 56-57 ‘when life history strategies differ among stages’ -unclear, rephrase
. Line 69 would it be possible, if data allow, to quantify the extent to which this kin factor affects social network position among helpers?
. Line 85-86 maybe the variation in the network connectivity within the life stages and its possible implications should be discussed to a much greater extent
. Line 303-305 it would be useful if the authors provide some information how such a drawback would potentially affect network metrics
. Line 325-330 From how it is phrased here I assume that the authors compared the network connectivity of helpers in 2016 that stayed helpers in 2017 to corresponding network connectivity of helpers in 2016 that transitioned to breeders in 2017. Only such comparisons would warrant a claim stating that ‘an individual’s social behavior as a helper may affect its probability of transitioning to a breeding position’. However, the Fig 3 that the authors refer to (as well as methods described in line 205-209 and reported results 250-255) seem to show completely different and irrelevant to the conclusions made by the authors comparisons. Please clarify.
. Line 342-346 An interesting point, it would be interesting to incorporate data on such within-life history stage variation in mass and social dominance rank into the current data set

.



.

---

## Round 0.2 · Minor Revisions

The two reviewers who reviewed your original submission have kindly provided feedback on your revision. Both commended the edits you have made and both recommended your article be accepted for publication. I agree with them.

One reviewer has a few outstanding comments that I ask you to address before I can accept your article for publication, however.

Additionally, I found the flow of the Introduction a little awkward and have made some suggested edits to the order in which you present the paragraphs to enhance the clarity of your arguments. Please see the attached annotated file. Acceptance is not contingent on you making these stylistic edits, but I suggest you consider them and the flow of your Introduction.

I look forward to receiving your revised article, and accepting your article for publication at that stage.

·

Basic reporting

See general comments

Experimental design

See general comments

Validity of the findings

See general comments

Additional comments

The authors have done an excellent job with revising their manuscript, and I applaud the efforts they have made to consider (and implement) the reviewers' comments. The manuscript is now much stronger, and clearly more robust. I have only two comments, both of which are relatively minor. These can be dealt with easily, and I do not need to review the manuscript again.

The main point is that the authors should ensure that they are putting in the right network data into the betweenness measure. In igraph, the betweenness measure (for some reason) interprets the weight of the edges as a cost or distance (see this thread on Twitter: https://twitter.com/mattjsilk/status/997029135548669957). Thus, the authors might want to follow the advice given about how to deal with this rather annoying issue. I'm sorry that I didn't pick this up in the first review.

Also in L62, I felt that two citations would make the introduction more complete:
Webber, Q. M., & Vander Wal, E. (2018). An evolutionary framework outlining the integration of individual social and spatial ecology. Journal of Animal Ecology, 87(1), 113-127.
He, P., et al. (2019). The role of habitat configuration in shaping social structure: a gap in studies of animal social complexity. Behavioral ecology and sociobiology, 73(1), 9.

It has been a pleasure to have had the opportunity to review this work.

Damien Farine
(I sign all of my reviews)

Reviewer 2 ·

Basic reporting

The authors have addressed all my suggestions and I have no further comments.

Experimental design

The authors have addressed all my suggestions and I have no further comments.

Validity of the findings

The authors have addressed all my suggestions and I have no further comments.

Additional comments

The authors have addressed all my suggestions and I have no further comments.

---

## Round 0.3 · accepted · Accept

Thank you for responding to the final outstanding queries.

It is my pleasure to accept your article for publication in PeerJ.